# Fallopian Tube Basal Stem Cells Reproducing the Epithelial Sheets In Vitro—Stem Cell of Fallopian Epithelium

**DOI:** 10.3390/biom10091270

**Published:** 2020-09-03

**Authors:** Maobi Zhu, Tomohiko Iwano, Sen Takeda

**Affiliations:** Department of Anatomy and Cell Biology, Faculty of Medicine, University of Yamanashi, 1110 Shimo-Kateau, Chuo, Yamanashi 409-3898, Japan; maobizhu@gmail.com

**Keywords:** oviduct, basal stem cell, cilia, differentiation, air–liquid interface (ALI), high-grade serous ovarian carcinoma (HGSC), ciliary motility, pig

## Abstract

The fallopian tube (FT) is an important reproductive organ in females. The luminal epithelium of the FT is composed of highly polarized secretory and ciliated cells. Recently, accumulating lines of evidence have suggested that the origin of high-grade serous ovarian carcinoma (HGSC) is fallopian tube epithelial cells (FTECs). Due to the lack of a high-fidelity model for FTECs in vitro, homeostasis, differentiation, as well as the transformation of FTECs are still enigmatic. In this study, we optimized the culture condition for the stable expansion of basal stem cells, as well as inducing differentiation of basal cells into polarized secretory and ciliated cells in the air–liquid interface (ALI) condition suitable for long-term culture. This storable culture method of FTECs provides a versatile platform for studying differentiation mechanisms, intercellular communication, and transformation to HGSC, as well as the physiological function of the FT in vitro.

## 1. Introduction

The fallopian tube (FT) is composed of three layers, and the innermost mucosa is a simple columnar epithelium consisting of basal, ciliated, and secretory cells [1]. The ciliated cells can facilitate the transport of gametes and mucus secreted by mucosal epithelium [2,3]. Accumulating evidence shows that high-grade serous ovarian carcinoma (HGSC) originates from epithelial cells lining the distal fallopian tube [4,5,6,7,8,9]. Currently, HGSC is the most common and aggressive subtype among ovarian carcinoma. However, the origin and molecular pathogenesis of HGSC remains unclear. HGSC is often characterized by the loss of ciliated cells concurrent with the expansion of proliferative secretory cells [10,11]. Because of the lack of a high-fidelity model for in vitro FTEC culture, the mechanisms involved in the homeostasis, differentiation, and transformation of FTEC remain enigmatic.

An immortalized FTECs cell line has been developed and this is a valuable tool for studying the cell biology of fallopian tube epithelium. It also offers us a stable tool with a long-term self-propagating cell population [12]. However, due to the immortalization process *per se*, the cellular profiles, especially those involved in mitosis and proliferation, have been somewhat changed. For example, the immortalized cells cannot be differentiated and do not give rise to a natural proportion of secretory and ciliated cells. From this standpoint, the immortalized cell line cannot substitute the primary culture from fresh tissues. An ideal culture model of FTECs should recapitulate the in vivo morphological and functional properties of cells along with the appropriate proportion of each cell type, in particular, the basal, ciliated, and secretory cells, to maintain the homeostasis of the epithelial sheet in long-term culture. The presence of stem cells has been found in the FT and concentrated in the distal end of the FT. The stem cells are the multipotent cells that differentiate into secretory or ciliated cells in the FT, which are indispensable for tissue regeneration in every menstrual cycle [8,13]. The presence of stem cells gives us the possibility to establish an in vitro culture of FTEC that recapitulates the biological environment in vivo. Looking back to the history of FTEC culture, various models of FTEC have been reported [14]. In early studies, FTECs were cultured by traditional methods, such as two-dimensional (2D) culture on a plastic dish with basal culture medium (e.g., DMEM medium with 10% FBS) without growth factors. This model did not induce the differentiation of FTECs or apicobasal polarity. Furthermore, these cell cultures developed a fibroblast-like morphology after long-term maintenance [15,16,17], suggesting the lack of stem cells that continuously supply the well-differentiated epithelial cells. Therefore, the early model did not represent the physiological function of FTECs and was not appropriate for studying FT function in vivo.

Air–liquid interface (ALI) culture was widely used for the culture of airway epithelial cells [18]. This model takes advantage of a semipermeable transwell membrane. Cells are grown on the upper chamber exposed to air in semi-wet conditions, while the bottom chamber is filled with differentiation medium to mimic the interaction of the epithelial basement membrane with capillaries. ALI culture is superior to submerged culture because it allows epithelial cells to differentiate in vitro [19]. This model was also used to support FTECs in a polarized state specific to the epithelial cells, and therefore recapitulate the physiological milieu of tissue in vivo [20,21,22,23,24]. However, this approach is inferior to three-dimensional (3D) organoid tissue. More recently, the 3D culture of FTECs was developed by the use of Matrigel. 3D cultures of FTECs allowed long-term maintenance and growth of organoid-like tissue containing both secretory and ciliated cells [25,26]. However, ALI culture still has advantages when compared with 3D culture, such as lower cost, reproducible differentiation, shorter incubation time, and more convenient analysis of ciliary function. 

From another standpoint, keeping the stemness that assures the differentiation of FTECs is often difficult in frozen stock. Furthermore, ingredients of culture media differ among each model, so the specific growth factors indispensable for long-term culture and maintenance of the stemness remain unknown. In most cases, the yield of primary basal FTECs depends on the isolation procedure from fresh tissue. It is difficult to maintain both a high growth rate and the stemness of FTECs over several passages. In addition, cryopreservation of FTECs in liquid nitrogen as a cell line to preserve optimal cellular profiles seems difficult. 

In lieu of usage of human fallopian tube tissue, establishing a stable and optimal culture model of porcine FTECs will benefit our understanding of optimizing the culture media, the number of expansion passages, storage conditions, and essential factors for differentiation. A stable high-fidelity system of culturing FTECs is also required to study the mechanisms of differentiation, intercellular communication, and transformation to HGSC, as well as the physiological function of FT.

In this study, we optimize the culture conditions for the stable expansion of basal stem cells from FT, as well as that for differentiation and long-term culture in ALI conditions. Moreover, cryopreservation is possible to maintain the stemness as well as differentiation capacity. This model may be useful to further study the biology of FTECs, such as the mechanisms of differentiation, epithelial–mesenchymal transition (EMT), and transformation to HGSC.

## 2. Methods and Materials

### 2.1. Cell Culture Medium

We defined three types of culture medium for FTECs at different stages. The “basal medium” used DMEM/Ham’s F-12 medium (042-30795, Wako, Chuo, Osaka, Japan) supplemented with 1% GlutaMAX (35050061, Thermo, Waltham, MA, USA), 2% B27 (17504-044, Thermo), 1 mM nicotinamide (72340, Sigma-Aldrich, St. Louis, MO, USA), 0.5 µM transforming growth factor beta (TGFβ) receptor kinase inhibitor IV (SB431542, Wako), and 10 ng/mL human epidermal growth factor (EGF; PHG0311, Thermo). This medium was used as the essential medium for the short-term culture of primary FTECs to maintain an undifferentiated status in ALI culture. The “expansion medium” was rich in growth factors that support the proliferation of FTECs and maintained the stemness during expansion. It was based on basal medium supplemented with 5 µM Rho-associated coiled-coil containing kinase (ROCK) inhibitor (030-24021, Wako), 100 ng/mL human fibroblast growth factor (FGF, 100-26, Peprotech, Rocky Hill, NJ, USA), 100 ng/mL human Noggin (120-10C, Peprotech), 50 ng/mL Wnt-3a (5036-WN-010, R&D systems, Minneapolis, MN, USA), and 125 ng/mL R-Spondin 2 (3266-RS-025, R&D systems). Finally, the basal medium with 2 ng/mL β-estradiol (E4389, Sigma-Aldrich) was used as “differentiation medium” for inducing differentiation in ALI culture.

### 2.2. Isolation and Culture of FTECs 

Fresh porcine fallopian tubes were obtained from healthy sows at a local slaughterhouse. The FTs were washed with phosphate-buffered saline (PBS) and opened longitudinally to expose the mucosal folds. Then, the whole FT was immersed in the digestion medium containing 100 U/mL collagenase type IV (17104-019, Thermo) and 10 µg/mL DNase I (9003-98-9, Sigma-Aldrich) for 90 min at 37 °C to release epithelial cells from the tissue. This procedure made the tissue sticky and hindered the release of epithelial cells into the medium. Scraping the epithelial surface was effective at improving the yield. Then, the tissue surfaces were rinsed with the same digestion medium for another 15 min at 37 °C to collect the epithelial sheet. If the medium remained sticky, the addition of further DNase I alleviated the situation. After digestion, the cell suspension was filtered with a cell strainer (Falcon, 100 µm pore 352360, New York, NY, USA) to remove the undigested aggregation, followed by centrifugation to collect cells. The collected cells were re-suspended in basal medium supplemented with 10% FBS, followed by seeding onto a collagen type I (PSC-1-200-100, Nippi, Tokyo, Japan)-coated plastic dish and then incubated at 37 °C in a 5% CO_2_-conditioned and humidified incubator. After the cells attached to the dish, the culture medium was replaced with serum-free medium. For long-term expansion, FTECs were co-cultured with proliferation-incompetent mouse embryo fibroblasts (MEFs) in the expansion medium. To prepare the inactivated MEFs, MEFs were incubated with 10 ug/mL Mitomycin C (Wako, 134-07911) for 2 h followed by washing with PBS 3 times. Usually, primary FTECs can be passaged at least five times and cryopreserved in liquid nitrogen.

For inducing the differentiation of FTECs, 1 × 10^5^ cells were seeded onto a collagen type I-coated transwell (0.4 µm pore size; Corning 3470, New York, NY, USA) and cultured in the basal medium with 10% FBS. After attachment, cultures were maintained with the serum-free basal medium until confluency. To induce differentiation in ALI cultures, the medium was removed from the upper chamber until the apical surface of the epithelial sheet was exposed to air, while the lower chamber continued to be filled with differentiation medium (ALI day 0). The medium was changed every 3 days.

### 2.3. Immunofluorescence

The FT and FTECs were rinsed twice with PBS and fixed in 4% paraformaldehyde (PFA) at room temperature (RT) for 10 min. Cells were washed three times with PBS, then permeabilized and blocked with 0.1% Triton-X 100 and 5% donkey serum in PBS for 30 min at RT. After blocking, samples were incubated with primary antibodies at 4 °C overnight. The following primary antibodies were used in this study: mouse anti-acetylated a-tubulin (6-11B-1; 1:500; Sigma-T6793), mouse anti-Foxj1 (14-9965; 1:200; Thermo), rabbit anti-p73 (1:200; ab40658, Abcam, Cambridge, UK), rabbit anti-Ki67 (NCL-ki67p, 1:500, Novocastra, Newcastle, UK), and rabbit anti-Pax8 (10336-1-AP, 1:400, Proteintech, Rosemont, IL, USA). Alexa Fluor 488/568-conjugated secondary antibodies (A21202/A10042, Thermo Fisher Scientific, Carlsbad, CA, USA) were used at a dilution of 1:200 at RT for 1 h. Nuclei were counterstained with DAPI (62248, Thermo Fisher Scientific). Finally, transwell membranes were mounted on glass slides using Diamond Antifade Mountant (P36961, Thermo Fisher Scientific). Images were viewed and taken by a confocal (Olympus FV-1000, Tokyo, Japan) or fluorescent (Olympus IX71) microscope.

### 2.4. Analysis of Proliferation Competency by 5-Ethynyl-2′-Deoxyuridine (EdU) Administration

The existence of precursor cells after induction of differentiation was shown by an EdU assay. Briefly, 5 µM EdU (C10639, Thermo Fisher Scientific) was added into the bottom chamber so that proliferation-competent cells would incorporate EdU during DNA synthesis. Detection of EdU was performed according to the manufacturer’s protocol.

### 2.5. Scanning Electron Microscopy (SEM)

Cell cultures were fixed in half Karnovsky’s solution that consisted of 2.5% glutaraldehyde and 2% PFA in 0.1 M cacodylate buffer. Then, the cells were rinsed three times with 10% sucrose containing cacodylate buffer, followed by post fixation in 1% osmium tetroxide for 30 min. The samples were thoroughly washed in distilled water and subsequently dehydrated in a graded series of ethanol solutions. After dehydration, samples were submerged in butyl alcohol medium for freeze-drying. The filters were cut from their support and mounted on stubs. After sputter coating with gold-palladium, the sample images were obtained with a JSM-6500 electron microscope (JEOL, Tokyo, Japan). 

### 2.6. Motility of Cilia

To capture a movie of cilia motility for the multi-ciliated cells (MCCs) from differentiated FTECs, the transwell membrane with cells was removed and placed on a glass slide with a coverslip set in the perfusion chamber. The chamber was mounted on a differential interference contrast (DIC) microscope (Olympus IX71, Tokyo, Japan) equipped with a high-speed camera (Prosilica GE-680, Allied Vision, Exton, PA, USA). A movie of ciliary motility was recorded and exported using TI Workbench software [27].

### 2.7. Statistical Analysis

Statistical analyses were performed using GraphPad Prism 6. Values were expressed as the means ± standard deviation (SD). Student’s *t*-test was used to compare the variation between two groups, while analysis of variance (ANOVA) was used to compare three or more groups. Statistical significance was assigned as * *p* < 0.05, ** *p* < 0.01, and *** *p* < 0.001.

## 3. Results

### 3.1. Identification and Stable Expansion of FTECs In Vitro

In view of topological variation in histological features from the isthmus to fimbria of fallopian tubes, the isthmus showed a thick muscular coat and simple mucosal folds, whereas both the ampulla and fimbria had complex mucosal folds with numerous epithelial cells (Figure 1A). Previous studies indicated that the distal end of the FT had a higher proliferation rate and self-renewal ability, with basal stem cells enriched in this region [8]. p73 is expressed in progenitor and ciliated cells and was shown to be necessary for ciliogenesis [28,29,30]. To identify the progenitor basal stem cells in the epithelial sheet, we stained the FT with anti-Ki67 and p73 antibodies. Consistent with previous studies, a high proportion of Ki67-positive cells were found in both the fimbria and ampulla, and the staining pattern suggested that the distal end of the FT was the source of stem cells (Figure 1A). p73 was expressed in progenitor cells and MCCs (Figure 1B).

The yield of harvested FTECs was higher in the ampulla and fimbria. Isolated epithelial cells were usually clump-like and maintained motile cilia, which propelled the clumps around the medium (Figure 1C). Because these clumps took a longer time to attach to the dish, medium replacement was avoided in the first two days. After two days, most clumps settled on the bottom of the dish and grew as colonies (Figure 1C). Cilia appeared to decrease during the cell expansion phase. Pax8 is specifically expressed in FTE [31,32]. About 95% isolated cells were Pax8-positive (Appendix A). This result suggests that most of the cells are FTECs, while a very small population belongs to other cell types represented by fibroblasts. For long-term expansion, the FTECs were co-cultured with MEFs (Figure 1C). Without MEFs, FTECs gradually detached from the dish. The colonies became confluent in a week. Feeder cells were removed by differential trypsinization, taking advantage of the higher sensitivity of feeder cells to trypsin than the strongly adherent epithelial cells (Appendix A). In the isolated state, primary FTECs co-cultured with MEFs in non-serum basal medium showed higher expression levels of p73 (Figure 1D). Taken together, these findings indicate that p73 is a useful marker for identifying primary FTECs with a differentiation property specific to progenitor cells. The stemness of primary FTECs was maintained in serum-free basal medium, which was sufficient to support the proliferation of cells while maintaining their identity in cell culture. 

To increase the number of FTECs, we tried to expand the FTECs with several passages while maintaining the proliferation and differentiation competency. For this purpose, we added growth factors to the basal medium to produce the expansion medium. When the FTECs co-cultured with MEFs were placed in the expansion medium, we observed rapid proliferation of cells after five days. The basal cell phenotype (p73 positivity) and proliferation competency (Ki67 positivity) were sustained over several passages (Figure 2A–C). On day five, the basal medium group still showed a relatively high proportion of p73- and Ki67-positive cells, even though this was slightly lower compared with those cultured in expansion medium. However, after 10 days, the difference became more obvious, and the proportion of p73- and Ki67-positive cells was significantly decreased, indicating a decreased growth rate (Figure 2A–C). Therefore, FTECs cultured in basal medium underwent a senescence accompanied by a loss of basal cell phenotype and a lower growth rate (Figure 2D). On the contrary, the expansion medium helped to sustain the multipotency and a higher growth rate of FTECs.

The study of FTE (fallopian tube epithelium) mainly relies on fresh cultures of FTECs, which have been regarded as fragile cells that would not tolerate cryopreservation to preserve their original properties. To test the proliferation competency and stemness of thawed FTECs, we investigated their cellular properties when co-cultured with MEFs in the expansion medium. While the expression of Ki67 had decreased at the fifth passage (P5) of expansion (Figure 3A,C), most cells still expressed high levels of p73 (Figure 3A,B). Moreover, FTECs also expressed low levels of Foxj1, which is a key transcriptional factor expressed just before ciliogenesis (Figure 3A). From our results, thawed FTECs were stable for at least five generations in the expansion medium. This demonstrated that primary FTECs can be cryopreserved in liquid nitrogen. The FTECs were stable and maintained their stemness when co-cultured with MEF cells in the expansion medium.

### 3.2. Establishing ALI Culture and Inducing Differentiation of FTECs

Considering the architectural similarity between airway epithelial cells and FTECs, we applied the ALI culture method to FTEC culture with optimization of culture medium for inducing the differentiation of FTECs. Expansion medium maintained the proliferation competency and stemness of FTECs. For inducing FTEC differentiation, we replaced the expansion medium with basal medium, removing some of the growth factors needed for maintaining stemness, such as Wnt3a, ROCK inhibitor, R-Spondin 2, Noggin, and FGF.

Because the FT is close to the ovary and the mucosal environment is modulated by two steroids during the menstrual cycle, estradiol (E2) is a potential factor inducing ciliogenesis in the FT. Our previous study showed that E2 is necessary and sufficient to induce ciliogenesis in FTECs [33]. When FTECs were grown in the presence of 2 ng/mL E2, MCCs were observed after approximately 10 days and showed full differentiation at day 15 (Figure 4A,B). Moreover, the differentiated FTECs shared similar morphological features with those FTE in vivo, which have both motile cilia and secretory cells with microvilli (Figure 4C).

The long-term culture of FTECs has been established with 3D Matrigel [25,26]. With this in mind, we examined whether ALI conditions were capable of sustaining FTECs in long-term culture. FTECs were cultured in basal medium with 2 ng/mL E2 for 27, 42, and 63 days under ALI conditions. FTECs survived and maintained differentiation competency with the presence of ciliated cells (Figure 4D). Moreover, these characteristics were sustained under these culture conditions for more than four months. We added EdU to the basal chamber for seven days before fixation and found that Edu-positive FTECs were present in the long-term cultures (Figure 4E). Taken together, the ALI culture model with basal media and E2 were sufficient for the long-term culture of FTECs. The FTECs maintained self-renewal and differentiation capacity, each shown by the presence of EdU-positive and ciliated cells, respectively. Furthermore, FTECs showed vigorous ciliary motility directly captured by a high-speed camera (Appendix A). This culture model provides a high-fidelity system that allowed FTECs to self-renew and differentiate into an epithelial sheet with physiological properties.

### 3.3. Defining the Effect of Growth Factors on the Differentiation Stage

Although the basal medium used in ALI culture was minimal in terms of the types of growth factors, there are still five components in this medium: B27, GlutaMAX, nicotinamide, TGFβ receptor kinase inhibitor IV (SB431542), and EGF. We defined the effect of each growth factor upon FTEC differentiation. In the serum-free basal medium, at least B27 is necessary to support cell growth. Therefore, we removed components one-by-one, except for B27, from the basal medium during the differentiation stage. We defined these modified media as 2 Mix (B27, GlutaMAX), 3 Mix (B27, GlutaMAX, nicotinamide), 4 Mix (B27, GlutaMAX, nicotinamide, SB431542), and 5 Mix (B27, GlutaMAX, nicotinamide, SB431542, EGF). 

Surprisingly, we found that only FTECs in 5 Mix showed a delay in ciliogenesis, whereas others showed early ciliogenesis on ALI culture day 5, and reached full differentiation state on ALI day 13 (Figure 5A). Because EGF is a mitogen that triggers cell proliferation, we added EdU to the medium for 48 h before fixation to examine cell proliferation in ALI culture. Contrary to our expectation, FTECs in 5 Mix did not show a higher proportion of EdU-positive cells, suggesting that EGF did not induce proliferation (Figure 5C). The proportion of EdU-positive cells appeared to be consistent with the number of ciliated cells, and EdU-positive cells were located near the ciliated cells. Except for the 5 Mix group, the other three groups showed a similar pattern of EdU incorporation (Figure 5A,B).

We predicted that EGF was not a key factor activating FTECs proliferation because of contact inhibition, at least in confluent cells on the transwell system. Instead, EGF may modulate the renewal and proliferation of intermediate precursor cells when their neighboring cells commit to MCCs. Assuming that the precursor cells next to differentiated MCCs will divide and replace the aging differentiated cells in long-term culture, it is reasonable that the groups without EGF showed a higher number of EdU-positive cells as the frequency of aging-differentiated cells increased. Because EGF inhibited cell differentiation and reduced cell renewal, we did not observe a high proportion of EdU-positive cells in the 5 Mix group. The other four factors in the media appeared to provide basic metabolic nutrients without affecting cell differentiation.

Considering that EGF is widely used in epithelial cell culture to maintain morphology and homeostasis in long-term cultures [25], we compared the morphology of FTECs in ALI culture with 4 Mix medium containing different concentrations of EGF. The number of ciliated cells was reduced with high concentration of EGF (10 ng/mL) at ALI days 7 and 15 (Figure 6). Highly polarized features of epithelial cells, such as a well-demarcated cell boundary, motile cilia, and microvilli, were observed in the groups supplemented with 2 and 10 ng/mL EGF. However, both ciliated and secretory cells tended to degenerate in the absence of EGF. In particular, secretory cells were more inclined to degenerate without EGF (Figure 6). Taken together, these findings indicate that EGF is essential for supporting long-term culture and maintaining the normal biological architecture of FT epithelial sheet. An optimal concentration of EGF is necessary for normal ciliogenesis in the FTE.

## 4. Discussion

In this study, we have established a primary culture model of FTECs which can recapitulate the in vivo epithelial sheet. Furthermore, this is a flexible model in which the natural course of differentiation step can be reproduced. This model will serve for studying the physiology and pathology of FTECs (Figure 7). The FT is adjacent to the ovary and its epithelium frequently renews itself during continuous exposure to inflammatory factors and ovarian hormones released every menstrual cycle. Basal stem cells are thought to be enriched in the distal end of the FT to meet the requirements of tissue renewal [8]. The staining pattern of Ki67 in the FT also demonstrated a high proliferation competency in the fimbria. Other key factors, such as p63 and p73, are more important in controlling the cell cycle and differentiation [34,35]. Previous studies reported that p73 was co-expressed with Foxj1 in basal cells of mice trachea, and was necessary for ependymal cell maturation, suggesting that p73 is needed for MCCs differentiation [28,36]. In p73 knockout mice, the FT showed defective ciliogenesis [28,30,37]. Thus, p73 and Ki67 have been shown to provide useful markers for identifying basal stem cells in FT.

While the FTE shares a similarity with tracheal epithelium (TE) in terms of ciliogenesis, we have considerably more knowledge on the biology and differentiation mechanisms of TE compared with that of the FT, as there is a well-established culture protocol for TE. In contrast, little is known about the biology and homeostasis of the FTE because of the lack of stable and systematic culture models in vitro. The FTECs are highly polarized cells that lose morphological features and function when cultured in submerged conditions. In this study, we established an in vitro protocol allowing the stable expansion and systematic differentiation of FTECs, covering the initial expansion, passaging, cryopreservation, and induction of differentiation steps. The most critical requirement is expanding the cell population while simultaneously preserving the stemness and differentiation competency. We developed a co-culture of FTECs with proliferation-incompetent MEFs in expansion medium that enhanced proliferation, as verified by Ki67 staining. The obtained cells remained stable after repeated passages on plastic dishes, and they maintained characteristics inherent to the original cells in situ, at least in terms of proliferation and differentiation. Moreover, the cells cryopreserved in liquid N_2_ can be used for long-term research, overcoming the shortage of human samples, and allowing repeated experiments with the same specimen. 

Recently, ALI culture was used for FTECs and it provided an excellent model to examine the function of FTECs [20,22]. This previous ALI culture model of FTECs relied on the fresh FT tissue and did not involve expansion or cryopreservation. Moreover, the past “ex vivo” culture model was incapable of self-renewal, provided only a month of culture, and lacked competency to be stably propagated for long-term use [14,20,23]. In our study, we have overcome the two limitations in the cell expansion and ALI culture stages, and have established a valuable new culture model for the expansion and differentiation of FTECs. This model can meet the needs of limited fresh tissue samples, and keep a long-term self-renewing cell population with more flexibility at a lower cost. 

Here, we have also improved the ALI culture conditions of FTECs. We used serum-free medium and minimal growth factors to induce FTEC differentiation, ensuring the stability and reproducibility of culture conditions, while eliminating the influence of many undefined factors and providing a new opportunity to study the mechanisms of FTEC differentiation. Indeed, we have successfully applied this culture model to the study of the molecular pathway in FTEC differentiation [33]. As summarized in Table 1, our method meets all the criteria that are essential prerequisites to have wide applications. In this regard, this model also allows investigation of the physiological responses of specific cell subpopulations, enabling us to better understand the mechanisms of tumor initiation and progression in FTECs leading to HGSC.

## Figures and Tables

**Figure 1 biomolecules-10-01270-f001:**
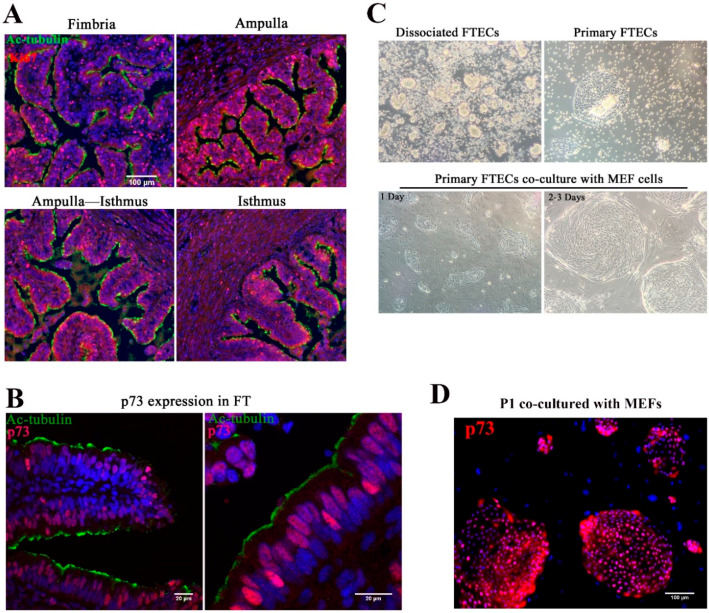
Isolation and characterization of primary fallopian tube epithelium cells (FTECs). (**A**) Immunostaining of porcine fallopian tube epithelium from fimbria to isthmus by anti-acetylated-tubulin (Ac-tubulin) and anti-Ki67 antibodies, counterstained with DAPI. Scale bars: 100 µm. (**B**) Immunostaining of porcine fallopian tube epithelium by anti-acetylated-tubulin, -p73, and DAPI. p73 was used as basal cell marker for FTECs. Scale bars: 20 µm. (**C**) FTECs released from the FT after digestion with collagenase type IV and DNase I. Mobile small clumps that exhibited cilia were observed in the medium. These clumps were the source of primary FTECs. The cell clumps attached to the dish after 2 days in the presence of 10% FBS. After reaching confluence, the FTECs were trypsinized and co-cultured with proliferation-incompetent feeder mouse embryo fibroblast (MEF) cells in serum-free expansion medium. The FTECs grew in a colony-like manner, which became bigger during culture and reached confluence in 1 week. (**D**) Primary FTECs co-cultured with MEF cells in basal medium also showed p73-positive colonies. Scale bar: 100 µm.

**Figure 2 biomolecules-10-01270-f002:**
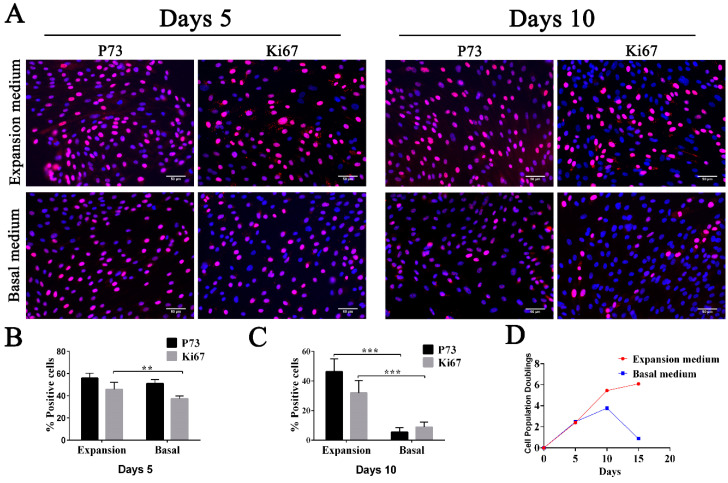
Characterization and proliferation rate of fallopian tube epithelium cells (FTECs) during expansion. (**A**) Primary culture of FTECs were stained by immunofluorescence with a marker for actively dividing cells (Ki67, red), a basal cell marker (p73, red), and DAPI (blue) 5 and 10 days after incubation in expansion or basal medium. Scale bars: 50 µm. (**B**,**C**) Quantitation of Ki67- and p73-positive cells at 5 days and 10 days of culture (analysis of variance (ANOVA) test, *n* = 7). (**D**) Doubling of population of primary cultured FTECs grown in expansion or basal medium over time. ** *p* < 0.01, and *** *p* < 0.001.

**Figure 3 biomolecules-10-01270-f003:**
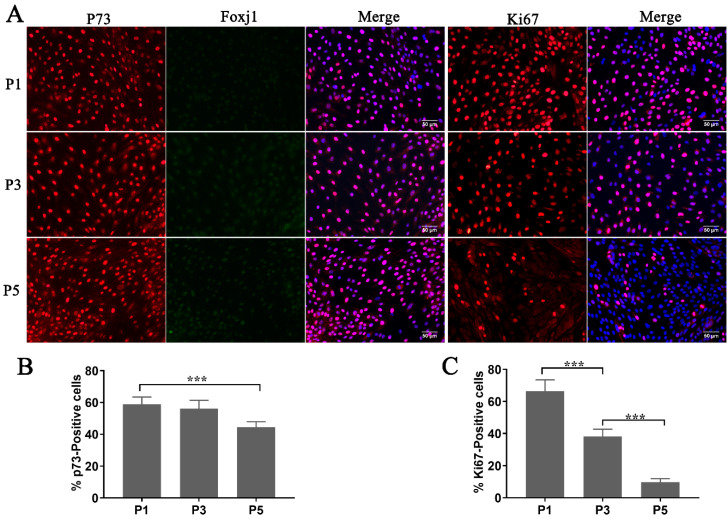
Evaluation of the differentiation competency and proliferation rate in cryopreserved fallopian tube epithelium cells (FTECs) during expansion. (**A**) FTECs at different numbers of passages were stained with molecular markers for progenitor (p73, red), ciliated (Foxj1, green), and mitotic (Ki67, red) cells. They were counter-stained with DAPI (blue) to visualize nuclei for cell counting. Scale bars: 50 µm. (**B**,**C**) Quantitation of Ki67- and p73-positive cells (ANOVA test, *n* = 7). *** *p* < 0.001.

**Figure 4 biomolecules-10-01270-f004:**
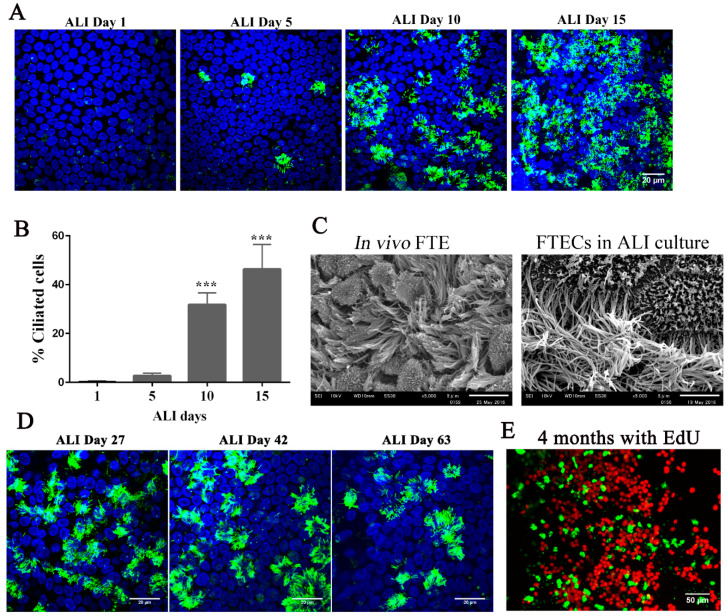
Inducing the differentiation of fallopian tube epithelium cells (FTECs) in air–liquid interface (ALI) culture. (**A**) Differentiation of FTECs was induced by adding 2 ng/mL estradiol (E2). Cells were stained with an anti-Ac-tubulin antibody (cilia, green) and DAPI (nuclei, blue) at ALI culture days 1, 5, 10, and 15. Scale bars: 20 µm. (**B**) Quantification of Ac-tubulin-positive cells (ANOVA test, n = 5, compared with ALI culture day 1 group). (**C**) Gross appearance of FTE tissue in vivo and differentiated FTECs cultured in ALI. (**D**) FTECs were thawed and cultured with 2 ng/mL E2 in basal medium. Cells on ALI culture days 27, 42, and 63 were stained with anti-Ac-tubulin antibody (green) and DAPI (blue). Scale bars: 20 µm. (**E**) FTECs were cultured with 2 ng/mL E2 in basal medium for 4 months. 10 µM Edu was added to the medium (bottom chamber) for 1 week, then cells were fixed and stained with anti-Ac-tubulin antibody (green) and DAPI (blue), and Edu-positive cells are shown in red. Scale bars: 50 µm. *** *p* < 0.001.

**Figure 5 biomolecules-10-01270-f005:**
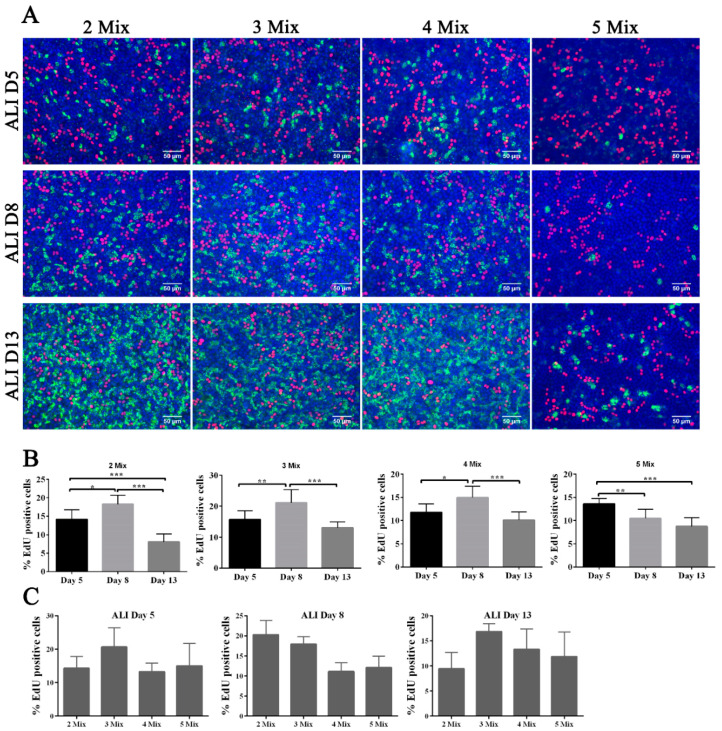
Effects of each factor in basal medium on fallopian tube epithelial cell (FTEC) differentiation and self-renewal. (**A**) FTECs were cultured in the different medium 2 Mix (B27, GlutaMAX), 3 Mix (B27, GlutaMAX, nicotinamide), 4 Mix (B27, GlutaMAX, nicotinamide, SB431542), and 5 Mix (B27, GlutaMAX, nicotinamide, SB431542, EGF), and exposed to ALI culture conditions. EdU (5 µM) was added to each group 48 h before the fixation on ALI culture days 5, 8, and 13. Fixed cells were stained for with anti-Ac-tubulin antibody (green), DAPI (blue), and processed for EdU staining (red). Scale bars: 50 µm. (**B**) Analysis of time-dependent EdU-positive cells in each culture medium using ALI culture conditions. (**C**) The number of EdU-positive cells were calculated in each medium on ALI culture days 5, 8, and 13. * *p* < 0.05, ** *p* < 0.01, and *** *p* < 0.001.

**Figure 6 biomolecules-10-01270-f006:**
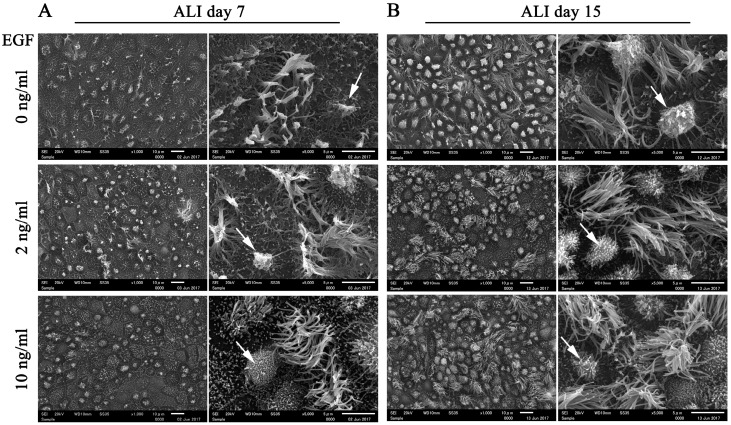
Effects of epidermal growth factor (EGF) on the morphology of differentiated cells. (**A**,**B**) FTECs were incubated with 4 Mix medium, supplied with different concentrations of EGF, as well as 2 ng/mL E2 to induce differentiation. Cells were fixed and observed by scanning electron microscopy (SEM) on ALI culture days 7 and 15. Scale bars: left panel, 10 µm; right panel, 5 µm.

**Figure 7 biomolecules-10-01270-f007:**
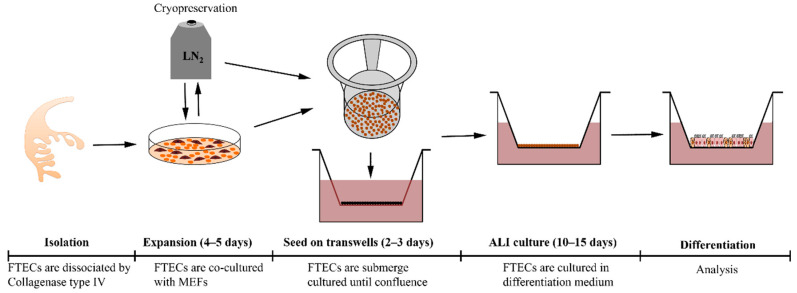
Overview of FTECs primary culture and strategy for inducing the multi-ciliated cells (MCCs) by ALI culture. FTECs were dissociated by collagenase IV and co-cultured with MEF cells to expand the colonies. After expansion, FTECs were seeded onto collagen I-coated porous supports and cultured with the basal medium. After reaching confluency, defined supplements were added to induce ciliogenesis. ALI culture conditions involved removing the medium from the top chamber so the liquid interface barely covered the apical surface of the epithelial sheet.

**Table 1 biomolecules-10-01270-t001:** Comparison of different culture systems of FTECs.

Species	Isolation	Culture Method	Key Supplements	Passage	Storage	Ciliogenesis	Reference
Porcine	Collagenase IV, DNAse I	Expansion: 2D co-culture with MEF cells, Differentiation: ALI culture	Basal medium, Expansion medium	Yes, 5 passages	Yes	Yes	This study
Human	Pronase, DNAse	ALI culture	2% USG	No	No	Yes	[20,24]
Human	Collagenase A	Co-culture with feeder cells, ALI culture	SCM-6F8 medium contained R-sponding1, Jagged-1, ROCK inhibitor, SB431542, nicotinamide	No	No	Yes	[21]
Porcine	Collagenase 1A	ALI culture	Conditioned medium	No	Yes	Yes	[22,23]
Human	Tissue fragment	Tissue culture on transwells	BSA, Fetuin B, Insulin, Transferrin, Selenium	No	No	/	[33]
Human	Collagenase I	3D culture	Wnt3A, RSPO1, GlutaMAX, B27, N2, EGF, Nicotinamide, Noggin, FGF, ROCK inhibitor, SB431542	Yes	No	Yes	[26]
Mouse	Collagenase A	3D culture	GlutaMAX, B27, EGF, Noggin, FGF, ROCK inhibitor, SB431542, Wnt3A, RSPO1	No	No	Yes	[25]
Human	Collagenase trypsin	2D culture	10% FBS, Insulin	Yes	No	No	[15]
Human	Brushed by Cytobrush	2D culture, 3D culture	15% FBS, EGF, Hydrocortisone, Insulin, Bovine pituitary extract	Yes	No	No	[17]
Human	Pancreatin, trypsin	2D culture	BSA, 2.5% FBS, 2.5% NuSerum	No	No	Little cilia	[16]

## Data Availability

The data that support the findings of this study are available from the corresponding author upon reasonable request.

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
