# Peer review of "Fallopian Tube Basal Stem Cells Reproducing the Epithelial Sheets In Vitro—Stem Cell of Fallopian Epithelium"

_biomolecules, 2020, doi:10.3390/biom10091270_

Round 1
Reviewer 1 Report
We do not have much comment on the work by Takeda’s laboratory. Their current work primarily describes the storable culture method for fallopian tube epithelial cells. We hope that the lab will soon have new and exciting functional studies using this new culture method.
Minor comments:
- Please clarify how many passages can the cells be expanded.
- Immortalized fallopian tube epithelial cells are available or can now be generated relatively easily [PMID: 22936217]. Immortalized cells allow the same batch of cells to be used, and reproducibility of the results will provide additional experimental rigors. Given the availability of immortalized cells, please also discuss how the storable primary culture is more superior than the immortalized cells.
Author Response
Reviewer 1
We do not have much comment on the work by Takeda’s laboratory. Their current work primarily describes the storable culture method for fallopian tube epithelial cells. We hope that the lab will soon have new and exciting functional studies using this new culture method.
Minor comments:
- Please clarify how many passages can the cells be expanded.
In this study, the primary FTECs can be passaged for 5-6 times in the expansion medium and mouse embryonic fibroblasts. In this condition, the cells still possess the capability of differentiation.
- Immortalized fallopian tube epithelial cells are available or can now be generated relatively easily [PMID: 22936217]. Immortalized cells allow the same batch of cells to be used, and the reproducibility of the results will provide additional experimental rigors. Given the availability of immortalized cells, please also discuss how the storable primary culture is more superior than the immortalized cells.
We have overlooked the document that reports the immortalized cell line. We appreciate his/her advice on this important point. We have modified the discussion section to incorporate this message of the immortalized cells on page 1 (introduction section, line35-41). While the immortalized cell line is advantageous in reproducibility and lower possibility of contamination, it is somewhat non-physiological, especially when it comes to the experiments that focus on the cell cycle, proliferation, and cancer biology. We have added these points to the discussion section.
Reviewer 2 Report
General comment
This study was conducted to optimize the culture conditions for the stable expansion of fallopian tube basal stem cells and improve the ALI culture to induce differentiation and long-term culture of isolated porcine oviduct cells. Establishment of a cost-efficient, reproducible culture system that is able to maintain the identity of primary fallopian tube epithelial cells could make a significant contribution to this ovarian cancer research. However, this study suffered from significant inadequacies, and the manuscript was poorly prepared. The comments are presented below:
Major comments
- The isolated cells were not well-characterized. The purity of the isolated cells needs to be verified.
- P73 was used as a basal cell biomarker. However, p73 is expressed in terminally differentiated MCCs and a subset of basal cells in the tracheal epithelium (PMID: 26947080). More specific biomarkers are required to identify basal cells.
- Figure 5. The connection between EdU incorporation and fallopian cell differentiation/self-renewal was not established.
- Table 1 is missing.
- Although the authors indicate, “FTECs cultured in basal medium underwent a senescence process accompanied by a loss of basal cell phenotype and a lower growth rate (Fig. 2D).” No data/evidence was presented to show that those are senescent cells.
- The authors indicate “the expansion medium maintained the stemness of FTECs and a higher growth rate.” The STEMNESS of these cells is not characterized.
- The manuscript was poorly prepared (e.g., table 1 is missing). It will be helpful if the authors have their manuscript checked by a native English speaker before resubmission.
Minor comments:
- What is MCC? Multiciliated cells?
- It is unclear why Figure 7 was cited before figure 4 in the main text.
Author Response
Reviewer 2
This study was conducted to optimize the culture conditions for the stable expansion of fallopian tube basal stem cells and improve the ALI culture to induce differentiation and long-term culture of isolated porcine oviduct cells. Establishment of a cost-efficient, reproducible culture system that is able to maintain the identity of primary fallopian tube epithelial cells could make a significant contribution to this ovarian cancer research. However, this study suffered from significant inadequacies, and the manuscript was poorly prepared. The comments are presented below:
Major comments
- The isolated cells were not well-characterized. The purity of the isolated cells needs to be verified.
Our method does not absolutely employ a specific procedure that targets on the isolation of FTECs. We took advantage of difference in the adhesiveness of FTECs and fibroblasts after trypsinization. However, most of the cells in each colony gave positive for p73 (figure 1D). Meanwhile, as one of the biological features of FTECs is the colony formation, we can eliminate the fibroblasts. After several passages, fibroblasts finally are lost. In addition, we also stained the cells by anti-Pax8 that is specifically expressed in FTE. Pax8-positive cells were over 95% (supplementary figure 1). This result suggests that most of the cells are FTECs, while a very small population belongs to other cell types represented by fibroblasts. This is a disadvantage of our method that makes ours different from the cell line.
- p73 was used as a basal cell biomarker. However, p73 is expressed in terminally differentiated MCCs and a subset of basal cells in the tracheal epithelium (PMID: 26947080). More specific biomarkers are required to identify basal cells.
According to your suggestion, we have stained the primary FTECs with anti-Pax8 antibody, which is a lineage-specific transcription factor expressed in the thyroid and FTECs. Pax8 was highly expressed in the basal cells [1,2]. About 95% isolated basal cells were Pax8 positive (supplementary figure 1).
- Figure 5. The connection between EdU incorporation and fallopian cell differentiation/self-renewal was not established.
We found EdU-positive cells in the differentiated epithelial sheets. This suggests that the EdU-positive cells still exist in the long-term culture. Taking the fact into account that EdU-incorporating cell, is indicative for mitosis, we suspected that EdU incorporated cells are basal cells that are capable of self-renewal. We will further clarify this point in future study.
- Table 1 is missing.
We are sorry for this disorganized manuscript. We put Table 1 on page 12, at the end of the discussion section.
- Although the authors indicate, “FTECs cultured in basal medium underwent a senescence process accompanied by a loss of basal cell phenotype and a lower growth rate (Fig. 2D).” No data/evidence was presented to show that those are senescent cells.
In figure 2D and figure 3C, we have shown the retarded growth rate accompanying Ki67-positivity. Although the basal cells of FTECs have stem-like properties, they do not divide themselves indefinitely. Therefore, we named it the intermediate stem cell. Since FTEC in later stage lose the proliferation competency that makes the cells proliferated, we assume that the cells undergo the senescent process. This is one of the important processes that need to be investigated.
- The authors indicate “the expansion medium maintained the stemness of FTECs and a higher growth rate.” The STEMNESS of these cells is not characterized.
We would like to thank this reviewer for his/her valuable suggestion. As has been suggested, it is necessary to identify them with specific molecular markers for stem cells. However, in our study, we showed two lines of evidence that the expansion medium is important for keeping the stemness of the FTECs. One is the high growth rate as shown in figure 2. Furthermore, we also have stained the cells with Foxj1 that is the marker of the ciliated cell lineage. The expression level of Foxj1 was increased during the ciliogenesis, which witnessed the differentiation of intermediate stem cell into ciliated cells. The expression of Foxj1 was very low during the expansion of FTECs. This is another testimony that FTECs still keep stemness. When we look at the expression of foxj1, it has been increasingly positive in P5 compared to P1, as shown in figure 3A. This is indirect evidence that the larger number of passages leads to loss of stemness.
- The manuscript was poorly prepared (e.g., table 1 is missing). It will be helpful if the authors have their manuscript checked by a native English speaker before resubmission.
We apologize for any inconvenience in reading this manuscript due to poor proofreading. We have fixed it by English editing. Table 1 is at the end of page 14.
Minor comments:
- What is MCC? Multiciliated cells?
Yes, “MCC” means multi-ciliated cell. We have included the non-abbreviated words on line 155, page 4.
- It is unclear why Figure 7 was cited before figure 4 in the main text.
Thank you for pointing out this incredible issue. We have added new content on page 13 and accordingly referred to figure 7. The initial citation of figure 7 has been deleted.
References
- Bowen, N.J.; Logani, S.; Dickerson, E.B.; Kapa, L.B.; Akhtar, M.; Benigno, B.B.; McDonald, J.F. Emerging roles for PAX8 in ovarian cancer and endosalpingeal development. Gynecol Oncol 2007, 104, 331-337, doi:10.1016/j.ygyno.2006.08.052.
- Rodgers, L.H.; E, O.h.; Young, A.N.; Burdette, J.E. Loss of PAX8 in high-grade serous ovarian cancer reduces cell survival despite unique modes of action in the fallopian tube and ovarian surface epithelium. Oncotarget 2016, 7, 32785-32795, doi:10.18632/oncotarget.9051.
Reviewer 3 Report
In their manuscript, “Fallopian tube basal stem cells reproducing the epithelial sheets in vitro”, Zhu et. al. describe a new model using porcine fallopian tube epithelial cells (FTECs) in an air-liquid interface (ALI) in vitro culture. The authors describe this model as an easy and effective method for maintaining basal stem cells which demonstrate appropriate differentiation. While easy and accurate models of FTE are important to study processes such as high grade serous ovarian cancer initiation, there are a few limitations which decrease the impact of this manuscript:
- During FTE isolation from porcine FTs, how are epithelial cells distinguished from other cell types (particularly from stromal cells such as fibroblasts or mesenchymal stem cells)?
- Why are MEFs used for feeder cells when you have access to porcine stromal/fibroblast cells from the initial sample? I system using cells from the same animal would be more appealing and more likely to be physiologically relevant.
- The justification for using an air-liquid interface (ALI) system is not clear. Unlike tracheal epithelium or skin epithelium, the FT is not exposed to air which questions the physiologic relevance of this system.
- The claim that p73 is a marker of progenitor cells in this work is not supported. The authors demonstrate p73 is expressed in a subset of cells and enriched with MEF co-culture but this does not mean it is a marker of progenitor cells. Further work demonstrating that p73 cells are the cells which give rise to both ciliated and secretory cells are needed to support this conclusion. Further, in general, stem/progenitor cells have a lower proliferation rate therefore it is unclear why p73 and ki67 would necessarily both increase to indicate increased stemness as proposed in figure 2. Proliferation rate (as indicated by Ki67) is distinct from the ability to self-renew and should be assessed separately.
- How are MEFs proliferation-incompetent? This is not described in the methods and it is unclear how pure the separation of FTE from MEFs is after differential trypsinization. How is the purity of the FTE confirmed?
Author Response
Reviewer 3
In their manuscript, “Fallopian tube basal stem cells reproducing the epithelial sheets in vitro”, Zhu et. al. describe a new model using porcine fallopian tube epithelial cells (FTECs) in an air-liquid interface (ALI) in vitro culture. The authors describe this model as an easy and effective method for maintaining basal stem cells which demonstrate appropriate differentiation. While easy and accurate models of FTE are important to study processes such as high grade serous ovarian cancer initiation, there are a few limitations which decrease the impact of this manuscript:
- During FTE isolation from porcine FTs, how are epithelial cells distinguished from other cell types (particularly from stromal cells such as fibroblasts or mesenchymal stem cells)?
We agree with his/her opinion that the preparation includes cell types other than epithelial cells (ECs). First, we took advantage of the differential adhesiveness of cell species to the substrate. While the epithelial cells prefer to form colonies bound each other by junctional complex and hard to be liberated by trypsinization, mesenchymal cells including the fibroblasts can be easily detached by enzymatic digestion. While somewhat a crude way, we have purified the ECs by passaging several times to eliminate the mesenchymal lineage. In addition, scraping the luminal surface of FT can minimize the contamination of other cell types than treating the cut FT of the whole wall with trypsin. In order to validate the purity of EC culture, we used the anti-p73 antibody. Most of the cells were p73 positive (figure 1D). Furthermore, we also employed the anti-pax8 antibody that is specifically expressed in FTECs [1,2]. Pax8-positive cells are about 95% (supplementary figure 1).
- Why are MEFs used for feeder cells when you have access to porcine stromal/fibroblast cells from the initial sample? I system using cells from the same animal would be more appealing and more likely to be physiologically relevant.
We completely agree with the reviewer’s idea that feeder cells have to be prepared from the same animal, from which FTECs have been prepared. However, it is almost impossible to prepare the porcine embryonic fibroblast (PEF), as we cannot access to the pig embryo. Therefore, we adopted the MEF that is widely used in the experiments. Considering the in vitro circumstances, we do not necessarily think too much about the immune system, which will abolish the PEF.
- The justification for using an air-liquid interface (ALI) system is not clear. Unlike tracheal epithelium or skin epithelium, the FT is not exposed to air which questions the physiologic relevance of this system.
As the reviewer has pointed out, FT is not absolutely exposed to the air as observed in airway epithelium. However, FT is a luminal organ, which is completely different from vascular endothelium that is completely soaked into blood and never communicates with the outer environment. While the thickness of mucous film over the epithelium differs to a certain degree between tracheal epithelium and FTEC, they basically share very similar architecture as the luminal organ that directly connects to the outer environment. FTE also possess highly polarized cells type, such as ciliated cells and secretory cells, both of which are found in the airway epithelium. This led us to employ ALI in this study. Under this condition in a two-compartment culture system called Transwell, FTECs have grown to polarized profiles. By scanning electron microscope (SEM), we also observed the differentiated FTECs with similar morphological profiles in vivo, where ciliated cells with motile cilia as well as secretory cells with microvilli (the new figure has been included in figure 4C) were identified.
- The claim that p73 is a marker of progenitor cells in this work is not supported. The authors demonstrate p73 is expressed in a subset of cells and enriched with MEF co-culture but this does not mean it is a marker of progenitor cells. Further work demonstrating that p73 cells are the cells which give rise to both ciliated and secretory cells are needed to support this conclusion. Further, in general, stem/progenitor cells have a lower proliferation rate therefore it is unclear why p73 and ki67 would necessarily both increase to indicate increased stemness as proposed in figure 2. Proliferation rate (as indicated by Ki67) is distinct from the ability to self-renew and should be assessed separately.
The important role of p73 in ciliogenesis has been well clarified in many studies, especially in the airway epithelial cells. In the p73 knockout mice, the fallopian tube showed defective ciliogenesis [3,4]. In this study, we also found p73 did express in the basal cells and ciliated cells in the fallopian tube tissue (figure 1B). Meanwhile, p73 is also highly expressed in almost all the primary basal FTECs, especially in the early passages (figure 1D, figure 1B). It means that the p73 can be used as a biomarker of the progenitor cells of ciliated cells.
We are sorry about the misunderstanding words in “stem cells”. In this study, we meant to defined the basal cells as a kind of adult stem cell but not the pluripotent stem cell. To evaluate the differentiation capacity of these basal cells during several passages, we use the p73 as a marker expressed in basal cells and ciliated cells. In this study, we also concern about the efficiency of expanding basal cells. So, we used Ki67 to optimized the culture condition that FTECs can be efficiently expanded for a longer time. In the expansion stage, we suppose the basal cells do not express pluripotent stem cell markers. The similar culture method of airway epithelial basal cells has verified this point [5]. We agree that the proliferation rate and self-renew and should be assessed separately. In figure 4E, EdU incorporated cells are found in basal cells that are capable of self-renewal in the long-term culture of FTECs not in the expansion stage indicated by Ki67.
- How are MEFs proliferation-incompetent? This is not described in the methods and it is unclear how pure the separation of FTE from MEFs is after differential trypsinization. How is the purity of the FTE confirmed?
We appreciate your valuable comments. To inactive the proliferation competency of MEF, we have treated the MEF with mitomycin C (10 ug/ml) for 2 hours. We also have included this information in the revised manuscript (page 3, line 118-119). Feeder cells were removed by differential trypsinization strategy taking advantage of the higher sensitivity of feeder cells to trypsin in comparison to strongly adherent epithelial cells (supplementary figure 2). It usually takes about 3-5 min to remove MEF from a dish, while FTEC remains to attach the dish over 10min. Therefore, we can remove nearly all the MEFs during the passage of the cell. We have added a figure in the supplementary data that shows the purification of the FTECs after the removal of MEF by trypsin.
References
- Bowen, N.J.; Logani, S.; Dickerson, E.B.; Kapa, L.B.; Akhtar, M.; Benigno, B.B.; McDonald, J.F. Emerging roles for PAX8 in ovarian cancer and endosalpingeal development. Gynecol Oncol 2007, 104, 331-337, doi:10.1016/j.ygyno.2006.08.052.
- Rodgers, L.H.; E, O.h.; Young, A.N.; Burdette, J.E. Loss of PAX8 in high-grade serous ovarian cancer reduces cell survival despite unique modes of action in the fallopian tube and ovarian surface epithelium. Oncotarget 2016, 7, 32785-32795, doi:10.18632/oncotarget.9051.
- Marshall, C.B.; Mays, D.J.; Beeler, J.S.; Rosenbluth, J.M.; Boyd, K.L.; Santos Guasch, G.L.; Shaver, T.M.; Tang, L.J.; Liu, Q.; Shyr, Y., et al. p73 Is Required for Multiciliogenesis and Regulates the Foxj1-Associated Gene Network. Cell Rep 2016, 14, 2289-2300, doi:10.1016/j.celrep.2016.02.035.
- Nemajerova, A.; Kramer, D.; Siller, S.S.; Herr, C.; Shomroni, O.; Pena, T.; Gallinas Suazo, C.; Glaser, K.; Wildung, M.; Steffen, H., et al. TAp73 is a central transcriptional regulator of airway multiciliogenesis. Genes Dev 2016, 30, 1300-1312, doi:10.1101/gad.279836.116.
- Butler, C.R.; Hynds, R.E.; Gowers, K.H.; Lee Ddo, H.; Brown, J.M.; Crowley, C.; Teixeira, V.H.; Smith, C.M.; Urbani, L.; Hamilton, N.J., et al. Rapid Expansion of Human Epithelial Stem Cells Suitable for Airway Tissue Engineering. Am J Respir Crit Care Med 2016, 194, 156-168, doi:10.1164/rccm.201507-1414OC.
Round 2
Reviewer 2 Report
The authors addressed some of concerns. They indicated that they will further clarify some concerns in their future study.
Author Response
Thank you for your valuable suggestions and comments during the whole process of review. In this version, we have already revised some mistakes of grammar, as well as the typos.
Reviewer 3 Report
Zhu et. al. have adequately addressed the concerns from prior review and edited their manuscript to reflect these changes. The authors comment on the potential of other cellular components in their system and use PAX8 expression as an additional marker to verify epithelial lineage. While the reviewer appreciates the usefulness of an air-liquid interface system to induce epithelial differentiation, the concerns regarding the physiologic relevance of this system remains however this is a minor concern. The manuscript requires minor editing to correct grammar and typos.
Author Response

(The authors gave the same response as above.)
